# Fine-Scale Coastal Storm Surge Disaster Vulnerability and Risk Assessment Model: A Case Study of Laizhou Bay, China

**Yueming Liu [1,2], Chen Lu [3], Xiaomei Yang [1,2,4,*], Zhihua Wang [1] and Bin Liu [1,2]**

1   State Key Laboratory of Resources and Environmental Information System, Institute of Geographic Sciences and Natural Resources Research, Chinese Academy of Sciences, Beijing 100101, China; liuym@lreis.ac.cn (Y.L.); zhwang@lreis.ac.cn (Z.W.); liub@lreis.ac.cn (B.L.)
2   University of Chinese Academy of Sciences, Beijing 100049, China
3   Land Satellite Remote Sensing Application Center, Ministry of Natural Resources of the People's Republic of China, Beijing 100048, China; luchen@lreis.ac.cn
4   Jiangsu Center for Collaborative Innovation in Geographical Information Resource Development and Application, Nanjing 210023, China
*   Correspondence: yangxm@lreis.ac.cn; Tel.: +86-10-6488-8955

**Abstract:** In the assessment of storm surge vulnerability, existing studies have often selected several types of disaster-bearing bodies and assessed their exposure. In reality, however, storm surges impact all types of disaster-bearing bodies in coastal and estuarine areas. Therefore, all types of disaster-bearing bodies exposed to storm surges should be considered when assessing exposure. In addition, geographical factors will also have an impact on the exposure of the affected bodies, and thus need to be fully considered. Hence, we propose a fine-scale coastal storm surge disaster vulnerability and risk assessment model. First, fine-scale land-use data were obtained based on high-resolution remote sensing images. Combined with natural geographic factors, such as the digital elevation model (DEM), slope, and distance to water, the exposure of the disaster-bearing bodies in each geographic unit of the coastal zone was comprehensively determined. A total of five indicators, such as the percentage of females and ratio of fishery products to the gross domestic product (GDP), were then selected to assess sensitivity. In addition, six indicators, including GDP and general public budget expenditure, were selected to assess adaptability. Utilizing the indicators constructed from exposure, sensitivity, and adaptability, a vulnerability assessment was performed in the coastal area of Laizhou Bay, China, which is at high risk from storm surges. Furthermore, the storm surge risk assessment was achieved in combination with storm water statistics. The results revealed that the Kenli District, Changyi City, and the Hanting District have a higher risk of storm surge and require more attention during storm surges. The storm surge vulnerability and risk assessment model proposed in this experiment fully considers the impact of the natural environment on the exposure indicators of the coastal zone's disaster-bearing bodies, and combines sensitivity, adaptability indicators, and storm water record data to conduct vulnerability and risk assessment. At the same time, the model proposed in this study can also realize multi-scale assessment of storm surge vulnerability and risk based on different scales of socioeconomic statistical data, which has the advantages of flexibility and ease of operation.

**Keywords:** storm surge; vulnerability; risk assessment; land-use; Laizhou Bay; coastal areas

## 1. Introduction

The intensity and frequency of storm surges are increasing under global climate change [1–5]. Casualties and economic losses from storm surges are determined in terms of two important aspects: hazard and vulnerability [6–8]. Natural hazards are highly variable and storm surge hazards are difficult to prevent and control using current scientific knowledge and technological capabilities [9]. Vulnerability to storm surge is the interaction between storm surges and a system (family, community, or country) [10]. A clear understanding of vulnerability can elucidate who and what are at risk from hazards, and how specific stresses and perturbations evolve into risks and impacts [11]. Therefore, it is necessary to conduct vulnerability assessments of storm surges in the coastal zone.

There are various definitions of vulnerability in the scientific literature [12–15]. Vulnerability was initially defined as the susceptibility to damage or injury during a given hazardous event [16–18]. In the 1970s and 1980s, research on vulnerability was limited to a specific hazard or the sensitivity (i.e., the ability to withstand impacts) of a specific type of disaster-bearing body [18,19]. More recently, exposure and adaptability (i.e., the ability to cope with a hazardous event and to recover from losses) have also become recognized as the basic components of vulnerability [18,20–22]. Vulnerability has thus evolved into a multidimensional structure. Against the backdrop of global climate change, vulnerability has become endowed with dynamic characteristics [10]. Vulnerability is not only affected by the current environment, but will also be determined in the future by changes to adaptation and mitigation actions as well as governance [6,23–29]. In this manuscript, vulnerability is defined as the combination of exposure, sensitivity, and adaptability. This will be further elaborated on in Section 3.

The basic method of assessing vulnerability to storm surge in large areas is to first construct a hierarchy of indicators, and then apply a comprehensive evaluation method from system science to conduct qualitative evaluation [30,31]. Alternatively, simple but effective formulas can be used to calculate indices to perform evaluation [32,33]. The assessment of vulnerability to storm surge has been carried out worldwide [33–40], including in China's coastal areas [31,32]. Storm surges primarily cause damage to coastal areas in the form of floods. Since floods are a more common type of disaster, scientists have performed more extensive research on this phenomenon [6,15,41–45]. Salman reviewed current approaches to modeling the potential impacts of climate change, population growth, increasing urbanization, and infrastructure decay on flood risk, and examined the current approaches to flood risk communication and public risk perception [43]. Paprotny used land use, population, and building information to calculate the exposure to floods from a broad range of scenarios in Poland [6]. Balica developed a Coastal City Flood Vulnerability Index (CCFVI) based on exposure, susceptibility, and resilience to coastal flooding, and conducted risk assessment for nine cities around the world [15]. Paprotny went a step further and began to study the compound floods of storm surge and inland runoff [44], investigating the dependence between the different drivers in different observational and modeled datasets, and utilizing gauge records and high-resolution outputs of climate reanalyses and hindcasts, as well as hydrodynamic models of European coasts and rivers.

In summary, it can be seen that the risk assessment method for coastal flood disasters is relatively mature, although the author found that there are still some deficiencies. First of all, in terms of exposure, the disaster-bearing body is located in the natural environment. The exposure of the disaster-bearing body is not fixed but will vary with the environment in which it is located. For example, when the same disaster-bearing body is located at a lower elevation or closer to a water source such as a river, it should have higher exposure. Therefore, it is not appropriate to assign the exposure of the disaster-bearing body to a fixed value, or to ignore the influence of the natural environment. Some researchers have introduced the digital elevation model (DEM) [6], which can alleviate this problem to a certain extent, but this study considered more factors. In addition to considering elevation, the effect of water source distance and slope was also taken into account. See Section 3 for details.

In addition, the vulnerability and risk assessment needs to be based on land use data. When a city develops rapidly and the land use in the coastal zone is updated frequently, the evaluation results will not be accurate if outdated land use data are used. This study solved this problem by adding

remote sensing data. Remote sensing displays the characteristics of both large-area simultaneous observation and high efficiency, and can provide data for the observation and prevention of various natural disasters [46]. There are currently more than 800 Earth observation and remote sensing satellites in operation [47], providing a large number of data sources for related research. The use of remote sensing data can obtain not only land use data at a specific point in time for vulnerability and risk assessment but can also utilize long-term remote sensing observation data in order to study changes in coastal city vulnerability. Furthermore, remote sensing can provide data such as the extent of inundated areas or conduct more comprehensive vulnerability and risk assessment studies via ocean satellite and meteorological satellite data. In this study, high-resolution optical remote sensing satellite images were used to obtain land cover data through image interpretation in order to achieve vulnerability and risk assessment based on remote sensing.

The main accomplishments of this study are as follows: (1) A storm surge vulnerability and risk assessment method combining remote sensing images is proposed; (2) When vulnerability assessment is performed, natural environmental factors are added to the exposure assessment; (3) Taking the coastal area of Laizhou Bay, Shandong Province, China as the research area, a storm surge vulnerability and risk assessment is conducted.

## 2. Study Area

The coastal area of Laizhou Bay is the region with the highest risk of storm surge in Bohai Bay, China [48,49]. In the past five years, the coastal area has endured 28 storm surges, causing serious threats to life and property [49]. Therefore, we utilized Laizhou Bay as an example to carry out storm surge vulnerability and risk assessment.

Laizhou Bay is located on the north side of the Shandong Peninsula, in the southern portion of the Bohai Sea. It is the largest bay in Shandong Province, China [49]. The geographical coordinates of the study area are 37°14′28.36″–37°57′3.42″N, 118°47′34.45″–120°37′32.45″E. There are four counties, three districts, and two county-level cities in the coastal area of Laizhou Bay. Dongying District and Kenli District belong to the prefecture-level city Dongying. Hanting District, Shouguang County, Guangrao County, and Changyi County belong to the prefecture-level city Weifang. Longkou City, Laizhou City, and Zhaoyuan County belong to the prefecture-level city Yantai. These nine administrative areas were used as assessment units. The study area is shown in Figure 1.

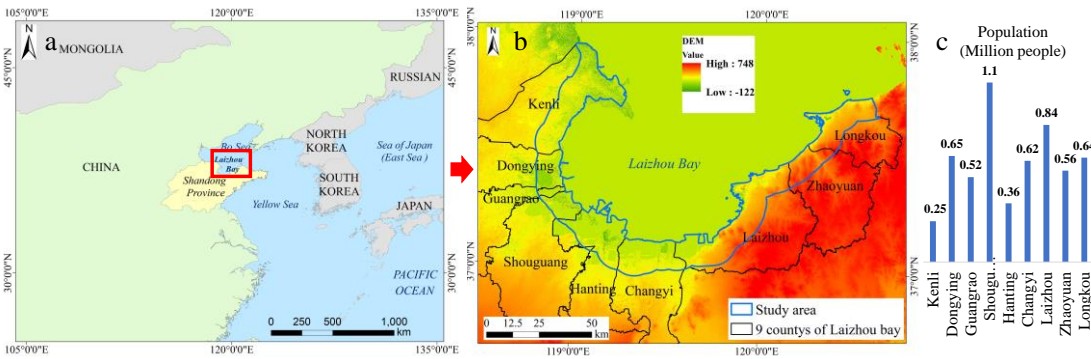

**Figure 1.** Map of the study area (**a**) Location; (**b**) Topography; (**c**) Population.

Along the coastline of Laizhou Bay, there are a large number of salt fields, fish ponds, buildings, and other disaster-bearing bodies. The elevation of 43% of the coastal area is <3 m. The coastal area has suffered from both typhoon storm surge and temperate storm surge. According to records in the literature [50], serious storm surge in the Laizhou Bay area can cause seawater backflushing, affecting areas up to 10 km inland from the shoreline. Therefore, the scope of this study was determined to extend from the coastline of Laizhou Bay to 10 km inland. In the coastal area of Laizhou Bay, most

of the disaster-bearing bodies are located on land, while few disaster-bearing bodies are on the sea. Therefore, the disaster-bearing bodies on the sea were excluded from the assessment.

The Laizhou Bay area frequently suffers from storm surge disasters. On average, this region experiences 30.2 storm surges with maximum water increases > 50 cm, 3.26 storm surges with maximum water increases > 150 cm, and 0.27 storm surges with maximum water increases > 250 cm each year [50]. According to observation station data, the maximum water increase record in this area is 358 cm [50]. Table 1 lists a sampling of disaster events and their associated damage in the Laizhou Bay area from 1990–2017 [50].

**Table 1.** Incomplete statistics of storm surge disaster events in Laizhou Bay (1990–2017).

| Date | Storm Surge Type | Maximum Storm Surge Increase (cm) | Main Disaster Data |
|---|---|---|---|
| 1992.08.31 | Typhoon 9216 | 304 | Seventy-six people died in Shandong Province, with a direct economic loss of more than 4 billion RMB |
| 1994.08.16 | Typhoon 9415 | 210 | No specific data |
| 1997.08.19 | Typhoon 9711 | 222 | Six people died in Shandong Province, with a direct economic loss of 720 million RMB |
| 2005.08.07 | Typhoon 0509 | 120 | Seven people died in Shandong Province, with a direct economic loss of 3.6 billion RMB |
| 2012.09.02 | Typhoon 1210 | 178 | Direct economic loss in Shandong Province: 1.6 billion RMB |
| 2003.10.11 | Cyclone + cold air | 307 | Direct economic loss in Shandong Province: 1 billion RMB |
| 2009.02.13 | Cyclone + cold air | 160 | No obvious disaster data |
| 2009.04.15 | Cyclone + cold air | 216 | No specific data |
| 2012.11.11 | Cyclone + cold air | 123 | Direct economic loss in Shandong Province: 149 million RMB |
| 2013.05.26 | Jianghuai cyclone | 120 | Direct economic loss in Shandong Province: 144 million RMB |
| 2014.10.11 | Cold air | 200 | Direct economic loss in Shandong Province: 29 million RMB |
| 2015.11.06 | Cold air | 185 | No specific data |
| 2016.10.21 | Cold air | 188 | Direct economic loss in Shandong Province: 89 million RMB |
| 2016.11.21 | Cold air | 180 | Direct economic loss in Shandong Province: 80 million RMB |
| 2017.10.09 | Cold air | 209 | Direct economic loss in Shandong Province: 6 million RMB |

## 3. Data and Method

### 3.1. Data Preparation

Prior to the vulnerability assessment, 21 remote sensing images with overall cloud cover of <1% were acquired over the coastal area of Laizhou Bay by the GF-2 sensor [51] (the satellite parameters are listed in Table 2). These 21 images captured from 16 February 2017–22 March 2017 cover the inland area up to 10 km from the coastline. The spatial resolutions of the panchromatic and multispectral images were 1 m and 4 m, respectively. The multispectral images were composed of color (RGB) and near-infrared (NIR) channels. In order to obtain the land-use distribution with high spatial precision, manual interpretation was conducted on the high-resolution remote sensing images. A total of 31 types of land use (listed in the second column of Table 3) were extracted from the remote sensing images via the land-use interpretation. Some land-use types containing disaster-bearing bodies of low economic value were not subdivided any further. For example, trees and grass were taken as one type of land use in the process of interpretation. Other important land-use types containing human

populations or disaster-bearing bodies with high economic value, however, were further subdivided. For instance, roads were subdivided into roads in built-up areas, railways, and highways.

**Table 2.** GF-2 satellite parameters.

| Parameter | 1-m Resolution Panchromatic/4-m Resolution Multispectral Camera | | |
|---|---|---|---|
| Spectral range | Pan | | 0.45–0.90 μm |
| | Multispectral | | 0.45–0.52 μm |
| | | | 0.52–0.59 μm |
| | | | 0.63–0.69 μm |
| | | | 0.77–0.89 μm |
| Resolution | Pan | | 1 m |
| | Multispectral | | 4 m |
| Gray level | | 16 bit | |
| Width | | 45 km | |
| Revisiting period (with side swing) | | 5 days | |
| Coverage repetitive period | | 69 days | |

**Table 3.** Land-use types and their exposure values.

| Exposure Value | Land-use Type |
|---|---|
| 0.1 | Bare land; grassland and woodland; under construction; river; lake; tidal flat |
| 0.2 | Arable land; orchard; reservoir |
| 0.3 | Pit pond |
| 0.4 | Park and green space; culture pond |
| 0.5 | Roads in built-up area |
| 0.6 | Fishing port; railway; highway; sports and recreation area |
| 0.7 | Industrial area; cargo port; land for public facilities; scenic area |
| 0.8 | Commercial and financial area; mining area of magnesium; salt field; press and publication area |
| 0.9 | Wholesale and retail area; accommodation and dining area |
| 1.0 | Rural residential area; urban residential area; campus; government and organization area; medical area |

The DEM data in the Laizhou Bay area were used to extract elevation and slope. Specifically, DEM data of the Shuttle Radar Topography Mission (SRTM) with a resolution of 30 m were used [52]. Covering the Laizhou Bay area, a total of nine DEM data scenes were utilized, with a row number range of n36–n38, e118–e120. The data source was the Computer Network Information Center of the Chinese Academy of Sciences (CAS).

Since the nine administrative units belong to the three prefecture-level cities—Dongying, Weifang, and Yantai—the 2018 statistical yearbooks of these prefecture-level cities were used as the main source of socioeconomic data [53]. The demographic data for each assessment unit were obtained from the 2010 population census of China [53].

According to the book "Bohai Sea Storm Surges and Disasters" published in 2019 [50], the largest storm surge database for the coastal waters of Laizhou Bay from 1960–2017 can be utilized as the basic data for the storm surge risk assessment of the coastal zone of Laizhou Bay.

*3.2. Remote Sensing Interpretation Method*

The remote sensing interpretation method used in this study was based on the combination of object-oriented segmentation and visual interpretation. The object-oriented segmentation was via segmentation of the remote sensing image into a patch object of a particular size, with the interpretation carried out at the scale of the patch rather than the pixel [54,55]. This method can effectively reduce salt-and-pepper noise and can form the land-use patch objects required by coastal research. A key step of using the object-oriented method is determining the segmentation scale. If the setting of the

segmentation scale is too small, the segmentation result is too fragmented, which is not conducive to the subsequent interpretation work and does not fully utilize the advantages of object-oriented segmentation. Meanwhile, if the segmentation scale is too large, it will lead to different terrain types being divided into the same patch, forming a "mixed patch," which is not conducive to interpretation. In this study, Wang's method was employed in the selection of the segmentation scale, and the segmentation parameters were adaptively selected [56]. The number of segmentation parameters was determined to be 90. After segmentation, the land-use types were classified based on visual interpretation. The visual interpretation method was selected because the results of the vulnerability and risk assessment in the follow-up study depended on the land classification results. It is difficult to obtain high-precision land-use products using the automatic classification method. Therefore, the visual interpretation method was chosen in order to ensure the accuracy of the land-use products as much as possible. The accuracy of the land-use products in this study was >95%, with the errors primarily resulting from the subtle differences between the patch segmentation edges and the actual edges of ground objects.

*3.3. Fine-Scale Coastal Storm Surge Disaster Vulnerability and Risk Assessment Model*

As described in the introduction, we know that the existing methods generally only consider the exposure of some of the disaster-bearing bodies when conducting storm surge vulnerability and risk assessment, and do not consider the impact of the natural environment on the disaster-bearing bodies. To address this issue, we propose a fine-scale coastal storm surge disaster vulnerability and risk assessment model, as shown in Figure 2. The risk assessment consists of two parts: the vulnerability of the disaster-bearing bodies and the hazard of the storm surge disaster. Vulnerability consists of three indicators: exposure, sensitivity, and adaptability. For fine-scale vulnerability and risk assessment, high-resolution remote sensing images were used to obtain data on land-use types in the study area, and the impact of natural environmental factors on the disaster-bearing bodies was taken into account in order to obtain a comprehensive exposure value. Sensitivity and adaptability indicators were then combined to achieve fine-scale vulnerability assessment. On this basis, in combination with storm surge hazard data, a fine-scale storm surge risk assessment can be achieved.

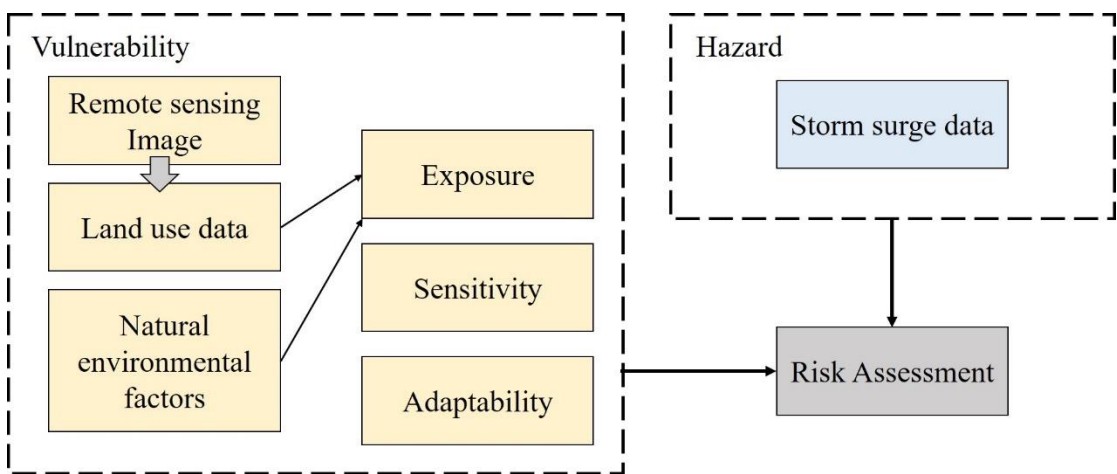

**Figure 2.** Fine-scale coastal storm surge disaster vulnerability and risk assessment model.

Utilizing an advantage of this method, the evaluation scale of this study was not fixed, but could be adjusted within a scale range. Through remote sensing data interpretation, we could obtain the fine scale (patch scale) land-use data, and, based on the different statistical units of social and economic data used, we could scale up, and then carry out vulnerability and risk assessment at the block scale, district and county scale, or city scale. In this study, we used the statistical data of districts and counties as the scale unit; thus, the results of this project are displayed at the scale of districts and counties.

There are many definitions of vulnerability available in the scientific literature [12–15], but there is no consensus as to the precise definition. In this study, the vulnerability definition presented by the Intergovernmental Panel on Climate Change (IPCC) was employed to construct an indicator system. In this case, vulnerability is the degree to which a system is susceptible to, or unable to cope with, adverse effects of climate change, including climate variability and extremes [23]. Vulnerability is a function of the character, magnitude, and rate of climate change and variation to which a system is exposed, its sensitivity, and its adaptive capacity [23]. Therefore, indicators were selected from three perspectives: exposure, sensitivity, and adaptability.

Exposure is defined as the nature, degree, and duration to which a system experiences environmental or sociopolitical stress [18]. In previous research, a subset of all of the disaster-bearing bodies was usually considered in exposure assessment. In the current study, the fine-scale land-use data obtained by visual interpretation were utilized to assess all of the types of disaster-bearing bodies exposed to storm surges in the coastal area of Laizhou Bay. Each of the 32 land-use types identified in this study correspond to one or several types of disaster-bearing bodies. For example, arable land use contains one type of disaster-bearing body—crops—whereas residential area land use primarily contains two types of disaster-bearing bodies: buildings and population. The areas of different land-use types were used as indicators, thus, all of the disaster-bearing bodies were included in the process of exposure assessment.

For all of the exposure indicators, the analytic hierarchy process (AHP [57]) was employed to conduct exposure assessments, which requires experts to judge the relative importance of each pair of any two indicators. If the areas of the 32 land-use types were selected as exposure indicators, experts would have to make 841 comparisons for a pair of indicators. This is a complex, time-consuming, and difficult task. Therefore, the 32 land-use types could not be directly used as indicators.

The State Oceanic Administration of China (2015) has issued a guideline for the risk assessment of storm surges [58]. The exposure values of different types of land use were stipulated in the guideline. A larger exposure value indicates that the storm surge will cause more casualties or economic losses. We applied the guideline and classified all land-use types into 10 groups. Each member of a land-use group has the same exposure value, and the exposure values of the 10 groups are 0.1, 0.2, 0.3, ... , 1. The land-use types and their exposure value are listed in Table 3. All of the 32 land-use types contained in the fine-scale data were assigned an exposure value.

Following this classification, the three indices of elevation, slope, and distance to water were selected to evaluate the impact of the natural environment on the exposure of the disaster-bearing bodies. The elevation and slope were extracted from the DEM data, and the distance to water was obtained by constructing a buffer zone for the water type in the land-use data of the study area. The impact of the three indicators on exposure is as follows: the lower the elevation, the more easily an area will be submerged by the tide, and the higher the exposure; the gentler the slope, the less conducive to drainage, and the higher the exposure; the closer an area is to the water, the more likely it is to be submerged and the higher the exposure. Exposure assignments were made to the three natural environmental indicators, and the results are listed in Table 4.

**Table 4.** Natural environment exposure indicators.

| Indicator | Effects on Exposure |
|-----------|---------------------|
| Elevation | Elevation of disaster-bearing body distribution. The lower the elevation, the more likely the disaster-bearing body is to be submerged and the higher its exposure. |
| Slope | Slope of disaster-bearing body distribution. The gentler the slope, the more unfavorable it is for water drainage, and the higher the exposure. |
| Distance to water | The closer the disaster-bearing body is to the water body, the greater the probability of being flooded and the higher the exposure. |

Different natural indicators exert different degrees of impact on the exposure of disaster-bearing bodies. The importance of each exposure index was determined through expert scoring, and the

weight of each index was comprehensively determined using the AHP method. Finally, the exposure and weight of each index were combined to form the final exposure of each land-use type tract.

Since both the sensitivity index and the adaptability index are socioeconomic statistical data, the statistics are based on districts. It is not possible to simply use the exposure value of the land-use type tract as a unit in the vulnerability assessment, and it is necessary to convert the exposure index to a district unit. Thus, the areas of the land-use type tracts with the same exposure values were employed as exposure indicators. The relative importance between two categories of land use was determined by comparing their exposure values. Let the exposure value of an indicator *S* be $W_s$, and the exposure value of an indicator *V* be $W_v$. Equation (1) was then employed to calculate the relative importance between indicator *S* and indicator *V*. Since the relative importance between any two indicators could be judged in an automated fashion, the exposure indicator weights could be calculated using the AHP method.

$$I_{sv} = \begin{cases} 8(w_s - w_v) + 1 \,, \ w_s > w_v \\ \frac{1}{8(w_v - w_s) + 1} \,, \ \ others \end{cases} \tag{1}$$

Sensitivity is the extent to which the system is susceptible to disasters. The sensitivities of population and economy were taken into account. The population sensitivity was assessed by indicators such as percentage of females, percentage of population under age 15, and percentage of population aged 65 and above. The ratio of fishery products to the gross domestic product (GDP) was selected as an indicator of economic sensitivity. The selected sensitivity indicators are listed in Table 5. The socioeconomic evaluation indicators in this study, including the sensitivity indicators in Table 5 and the adaptability indicators in Table 6, were ultimately determined by the relevant expert group members in the CAS Earth Big Data Science Project, as listed in the "Funding" section at the end of this manuscript, following the screening and evaluation of many available indicators.

**Table 5.** Sensitivity indicators and their effects on adaptability.

| Indicator | Effect on Sensitivity |
|---|---|
| Percentage of females | The physical strength of women is less than that of men. Thus, women are more easily affected by storm surges than men during a disaster event. |
| Percentage of population under age 15 | Children under the age of 15 are more vulnerable to storm surge disasters than adults. In China, children enter school at the age of six, receive nine years of compulsory education, and graduate at the age of 15. Compared with adults, it may be more difficult for children under the age of 15 to take appropriate measures to protect themselves in the face of sudden disasters. Hence, they have greater sensitivity. |
| Percentage of population aged 65 and above | Due to the decline in bodily functions, older people are more vulnerable to storm surge disasters than the young. |
| Percentage of population with junior, secondary, and lower education | People with less education tend to have lower incomes and possess fewer resources. Therefore, those with less education are more sensitive to storm surges. |
| Ratio of fishery products to gross domestic product (GDP) | An economic system with a large amount of fishery production is more susceptible to storm surge hazards. |

Adaptability is the ability of a system (family, community, region, or country) to cope, manage, or adjust to climate change, stress, disasters, and risks or opportunities [29]. Urban disposable income per capita and rural disposable income per capita were selected as indicators to assess the adaptability of the population. In addition, the adaptability of the social environment in which people live and work was also considered. General public budget expenditures, GDP, number of medical institutions, and other indicators were selected, as listed in Table 6.

**Table 6.** Adaptability indicators and their effects on adaptability.

| Indicator | Effect on Adaptability |
|---|---|
| General public budget expenditure | A region with higher public budget expenditure is expected to make more investments in disaster management. Therefore, the adaptability of such a region is higher than regions with less public budget expenditure. |
| GDP | A region with higher GDP tends to have more public budget expenditure, as well as higher levels of medical care, technology, and social security. All such advantages will increase the adaptability of such a region. |
| Urban disposable income per capita | If storm surge disasters cause damage to the property of urban residents, a higher income can ensure that the residents repair the damage faster. |
| Rural disposable income per capita | Rural residents who have higher income can repair the damage caused by storm surges faster. |
| Number of hospital medical staff | A higher quality of medical care protects human health before disasters and can treat more injured people during and after disasters. The higher quality of medical care thus increases adaptability of a coastal area. The number of hospital medical staff is a useful indicator for assessing medical care. |
| Number of medical institutions | The number of medical institutions is also an important indicator for assessing medical care. |

After constructing the indicator system of vulnerability assessment, the AHP method and the entropy method [59] were integrated to calculate the weights of each indicator. First, the AHP method was applied, and five experts (comprising two experts in the field of marine economy, one expert in the field of physical geography, and two experts in the field of geography information systems) were invited to rate the selected indicators. Since the relative importance of the exposure indicators was obtained by automated calculation, the experts only needed to conduct pairwise comparisons on the indicators of sensitivity and adaptability. The consistency test revealed that the consistency ratio was 0.014, and thus < 0.1, confirming the consistency of the five comparison matrices. Therefore, the weights generated by the AHP method were reliable. Subsequently, the entropy method was used to calculate the weights. In order to make full use of the expert knowledge and the characteristics of the data itself, the weights calculated by the above two methods were combined by Equation (2):

$$w_i = \frac{u_i v_i}{\sum_{j=i}^{n} u_j v_j} , \tag{2}$$

where $u_i$ is the weight of the indicator $i$ calculated by the AHP method, $v_i$ is the weight of the indicator $i$ calculated by the entropy method, and $w_i$ is the integrated weight of the indicator $i$ based on the AHP and entropy method.

After obtaining the vulnerability of the disaster-bearing bodies, the vulnerability was graded using the equal interval method in order to obtain the vulnerability level of each district. At the same time, the storm surge maximum water increase data were regarded as a hazard index, and were classified to obtain the storm surge hazard level of each district. Finally, through the correspondence between the vulnerability of the disaster-bearing bodies and the storm surge hazards from the storm surge risk assessment in the "Guideline for risk assessment and zoning of storm surge disasters" by the State Oceanic Administration of China [58], the final risk assessment results were determined, as shown in Table 7.

**Table 7.** Correspondence between vulnerability and hazard in risk assessment.

| | | Vulnerability | | | |
|---|---|---|---|---|---|
| | | Very Low (Level IV) Range [0.1,0.3] | Low (Level III) Range (0.3,0.5] | High (Level II) Range (0.5,0.8] | Very high (Level I) Range (0.8,1] |
| Hazard | Very low (Level IV) | Very low risk (Level IV) | Very low risk (Level IV) | Low risk (Level III) | Low risk (Level III) |
| | Low (Level III) | Very low risk (Level IV) | Low risk (Level III) | High risk (Level II) | High risk (Level II) |
| | High (Level II) | Low risk (Level III) | High risk (Level II) | High risk (Level II) | Very high risk (Level I) |
| | Very high (Level I) | Low risk (Level III) | High risk (Level II) | Very high risk (Level I) | Very high risk (Level I) |

## 4. Result

The land-use types in the Laizhou Bay area were obtained through visual interpretation (Figure 3), and the exposure values were assigned to the land-use types in the coastal zone of Laizhou Bay based on the corresponding relationship in Table 3. The results are shown in Figure 4a.

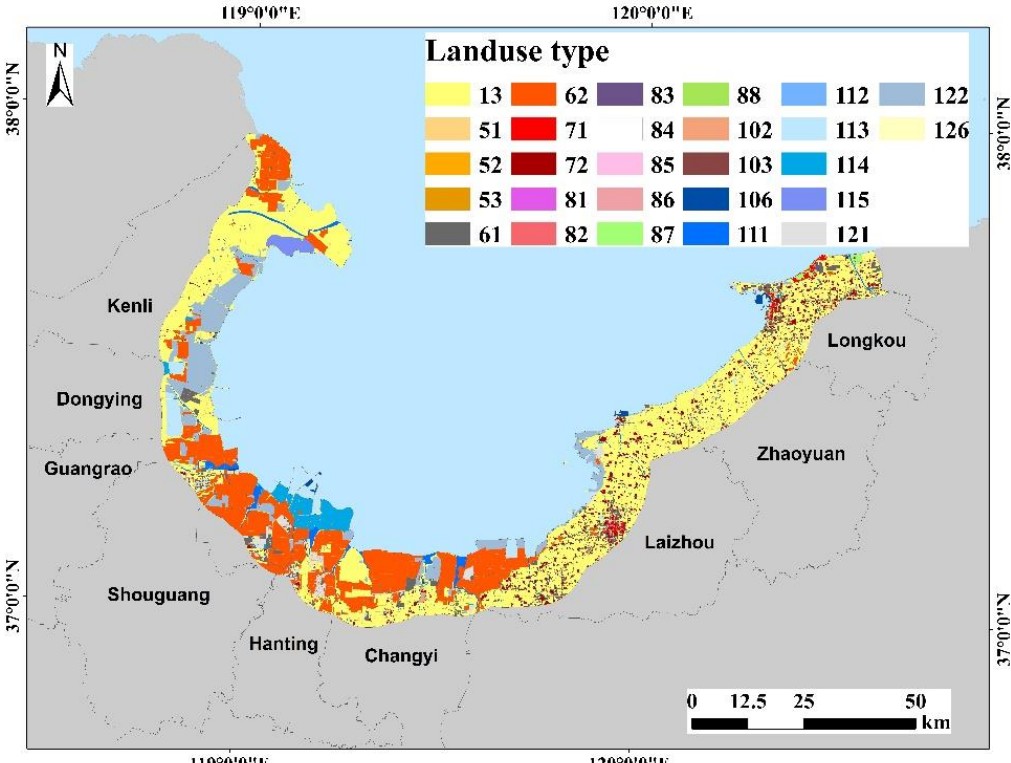

**Figure 3.** Interpretation results of land use in Laizhou Bay. Land-use types: 13 = arable land; 51 = wholesale and retail area; 52 = accommodation and dining area; 53 = commercial and financial area; 61 = industrial area; 62 = magnesium mining area; 71 = urban residential area; 72 = rural residential area; 81 = government and organization area; 82 = press and publication area; 83 = campus; 84 = medical area; 85 = sports and recreation area; 86 = land for public facilities; 87 = park and green space; 88 = scenic area; 102 = highway; 103 = road in built-up area; 106 = cargo port; 111 = river; 112 = lake; 113 = reservoir; 114 = pit pond; 115 = tidal flat; 121 = under construction; 122 = salt field; 126 = bare land.

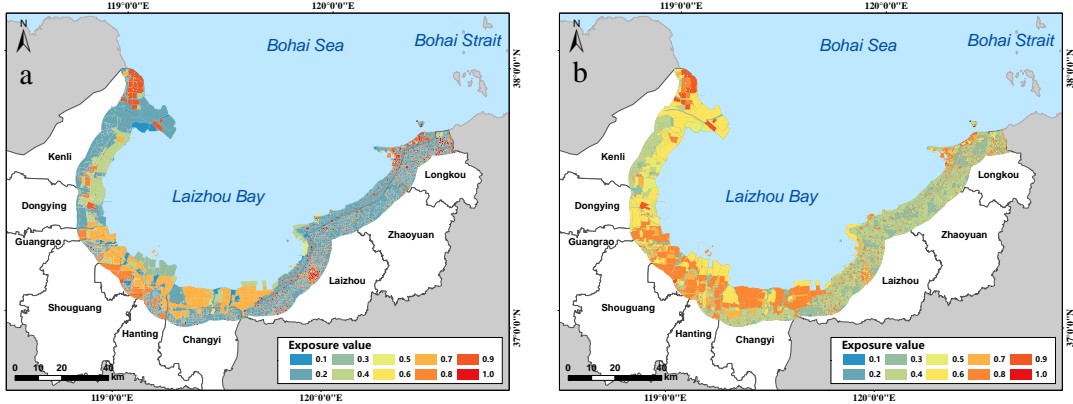

**Figure 4.** Exposure of the coastal zone of Laizhou Bay: (**a**) Land-use type exposure; (**b**) Exposure after considering the natural environment.

Taking into account the impact of the natural environment on the exposure of disaster-bearing bodies, the experts' scores for the exposure of the disaster-bearing bodies were as follows: land-use type exposure, 9; elevation, 3, slope, 2; and distance to water 4. The weights were calculated using AHP, as listed in Table 8.

**Table 8.** Exposure indicators and weights.

| Indicator | Judgment Criterion | Exposure Value | Experts' Score | Weight |
|---|---|---|---|---|
| Land-use type | Land-use type | Land-use type exposure | 9 | 0.5 |
| Evaluation | <0.5 m | 1 | | |
| | 0.5–1 m | 0.6 | 3 | 0.167 |
| | >1 m | 0.2 | | |
| Slope | 0–6° | 1 | | |
| | 6–20° | 0.6 | 2 | 0.111 |
| | >20° | 0.2 | | |
| Distance to water | ≤0.5 km | 1 | | |
| | 0.5–1 km | 0.8 | | |
| | 1–2 km | 0.6 | 4 | 0.222 |
| | 2–5 km | 0.4 | | |
| | >5 km | 0.2 | | |

Using the weighting calculation for each indicator, the exposure of the disaster-bearing bodies in the coastal zone of Laizhou Bay considering the natural environment impact was obtained, as shown in Figure 4b.

It can be seen from Figures 4 and 5 that after considering the natural environment factors, the exposures of the land-use types changed significantly. The areas of low-exposure and high-exposure land-use types decreased significantly, while the area of moderate exposure increased to a certain extent after adding the impact of the natural environment. The main reason for these changes is that before considering the impact of the natural environment, the exposure is only related to the type of land use, and the areas of different exposures directly reflect the areas of the corresponding land use type. When considering the impact of the natural environment, however, exposure becomes a comprehensive indicator that includes both land-use type and environmental impact. Only for cases in which land-use types and the environment simultaneously have low or high exposure will the final result have either low or high exposure; otherwise, it is more likely to exhibit moderate exposure. For example, when the environmental impact is not considered, the land-use type with an exposure of 0.2 in the Laizhou Bay area covers a large area, which is mainly due to the large amount of arable land. Most of the arable land in the Laizhou Bay area, however, is located in low-lying and gentle slope regions, and the natural

environment is more exposed. Therefore, after taking into consideration the impact of the natural environment, the comprehensive exposure increases. Ultimately, the area of the disaster-bearing bodies with an exposure of 0.2 was greatly reduced. In summary, after considering the impact of the natural environment, the conditions that produce low exposure and high exposure become more severe, and in this case, attention needs to be focused on features that still exhibit high exposure.

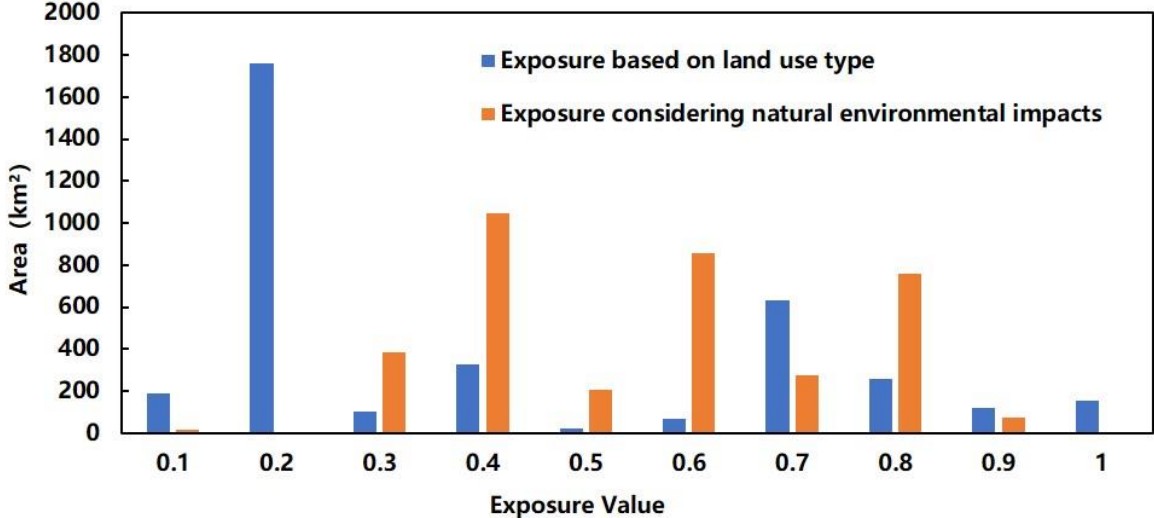

**Figure 5.** Areas of different exposure features in the coastal zone of Laizhou Bay before and after considering natural environmental impacts.

Based on the results of the exposure assessment that considered the natural environment, the area of land-use type with an exposure of 0.1–1.0 was calculated on a district basis, and combined with the economic statistical data obtained from the relevant statistical yearbook. The specific values of the exposure, sensitivity, and adaptability indicators were then obtained for each district of Laizhou Bay, as listed in Table 9.

The AHP method and entropy method were used to calculate the weights of each indicator, and Equation (2) was then utilized to combine the weights calculated by these methods.

The integrated weights of all of the indicators are presented in Table 10. Exposure and sensitivity contributed more than one-third of the total weight, and adaptation contributed less to vulnerability than exposure and sensitivity.

After generating the weights of all of the indicators, the vulnerability levels of the nine assessment units were calculated. The results of the exposure indices, sensitivity indices, adaptability indices, and vulnerability indices are illustrated in Figure 6. In the coastal area of Laizhou Bay, the nine assessment units ranked from high to low vulnerability were as follows: Kenli District, Longkou City, Hanting District, Changyi City, Laizhou City, Guangrao County, Dongying District, Zhaoyuan City, and Shouguang City.

**Table 9.** Specific values of vulnerability indicators in Laizhou Bay.

| Indicator | | | Kenli | Dongying | Guangrao | Shouguang | Hanting | Changyi | Laizhou | Zhaoyuan | Longkou |
|---|---|---|---|---|---|---|---|---|---|---|---|
| Exposure | Area of land-use with exposure values ranging from 0.1–1 (km$^2$) | 0.1 | 0.58 | 0.02 | 0.00 | 0.09 | 0.60 | 1.14 | 6.51 | 1.80 | 5.12 |
| | | 0.2 | 0.16 | 0.00 | 0.00 | 0.11 | 0.14 | 0.35 | 3.25 | 1.12 | 0.39 |
| | | 0.3 | 36.63 | 7.36 | 0.00 | 10.01 | 22.45 | 18.56 | 164.97 | 38.58 | 85.85 |
| | | 0.4 | 162.50 | 13.99 | 0.01 | 5.73 | 42.45 | 129.73 | 381.47 | 149.09 | 164.26 |
| | | 0.5 | 105.89 | 14.51 | 0.47 | 11.81 | 22.60 | 4.22 | 22.23 | 2.91 | 21.42 |
| | | 0.6 | 350.13 | 132.43 | 4.80 | 34.33 | 126.72 | 81.90 | 85.59 | 7.86 | 32.19 |
| | | 0.7 | 22.02 | 0.70 | 6.32 | 23.90 | 45.04 | 39.96 | 76.22 | 9.12 | 53.33 |
| | | 0.8 | 43.55 | 23.45 | 30.86 | 87.63 | 208.78 | 156.59 | 158.10 | 7.41 | 44.33 |
| | | 0.9 | 63.69 | 8.90 | 0.00 | 0.15 | 2.43 | 0.00 | 0.04 | 0.00 | 0.27 |
| | | 1.0 | 0.10 | 0.01 | 0.00 | 0.00 | 0.26 | 0.00 | 0.04 | 0.00 | 1.11 |
| Sensitivity | Percentage of females (%) | | 49.46 | 49.23 | 49.10 | 49.05 | 48.76 | 49.79 | 49.25 | 49.67 | 49.44 |
| | Percentage of population under age 15 (%) | | 15.57 | 15.16 | 16.28 | 15.03 | 14.69 | 14.80 | 10.98 | 12.32 | 11.43 |
| | Percentage of population aged 65 and above (%) | | 9.52 | 7.50 | 10.14 | 10.05 | 10.82 | 11.79 | 13.75 | 12.80 | 10.21 |
| | Percentage of population with junior, secondary, and lower education (%) | | 75.16 | 51.76 | 77.43 | 76.96 | 77.47 | 79.33 | 79.06 | 73.92 | 74.23 |
| Adaptability | Ratio of fishery products to GDP (%) | | 5.54 | 1.88 | 1.68 | 4.16 | 0.1 | 5.44 | 2.85 | 0.75 | 0.42 |
| | General public budget expenditure (billion yuan) | | 2.88 | 3.18 | 4.99 | 9.55 | 2.47 | 4.06 | 6.18 | 5.71 | 9.47 |
| | GDP (billion yuan) | | 45.48 | 49.97 | 86.92 | 86.67 | 23.34 | 44.29 | 76.93 | 74.01 | 119.09 |
| | Urban disposable income per capita (yuan) | | 38341 | 45394 | 40077 | 37606 | 34557 | 33693 | 42027 | 42181 | 45013 |
| | Rural disposable income per capita (yuan) | | 15605 | 18634 | 18681 | 19249 | 17312 | 17662 | 19557 | 19755 | 20554 |
| | Number of hospital medical staff | | 1069 | 9449 | 3075 | 8908 | 3202 | 4036 | 7665 | 3473 | 3948 |
| | Number of medical institutions | | 282 | 469 | 397 | 694 | 475 | 565 | 898 | 391 | 435 |

**Table 10.** Weights of indicators for assessing vulnerability.

| Vulnerability Dimension (Expert Score: 1–9) | Indicator | Expert Score | Weight |
|---|---|---|---|
| Exposure (9) | Areas of land-use with the exposure value of 0.1 | No expert scoring required; determined by formula (1) | 0.010 |
| | Areas of land-use with the exposure value of 0.2 | | 0.015 |
| | Areas of land-use with the exposure value of 0.3 | | 0.013 |
| | Areas of land-use with the exposure value of 0.4 | | 0.015 |
| | Areas of land-use with the exposure value of 0.5 | | 0.028 |
| | Areas of land-use with the exposure value of 0.6 | | 0.032 |
| | Areas of land-use with the exposure value of 0.7 | | 0.027 |
| | Areas of land-use with the exposure value of 0.8 | | 0.045 |
| | Areas of land-use with the exposure value of 0.9 | | 0.242 |
| | Areas of land-use with the exposure value of 1.0 | | 0.263 |
| Sensitivity (7) | Percentage of females | 7 | 0.023 |
| | Percentage of population under age 15 | 7 | 0.029 |
| | Percentage of population aged 65 and above | 6 | 0.018 |
| | Percentage of population with junior, secondary, and lower education | 5 | 0.010 |
| | Ratio of fishery products to GDP | 9 | 0.055 |
| Adaptability (9) | General public budget expenditure | 5 | 0.044 |
| | GDP | 1 | 0.006 |
| | Urban disposable income per capita | 3 | 0.031 |
| | Rural disposable income per capita | 3 | 0.027 |
| | Number of hospital medical staff | 5 | 0.043 |
| | Number of medical institutions | 5 | 0.025 |

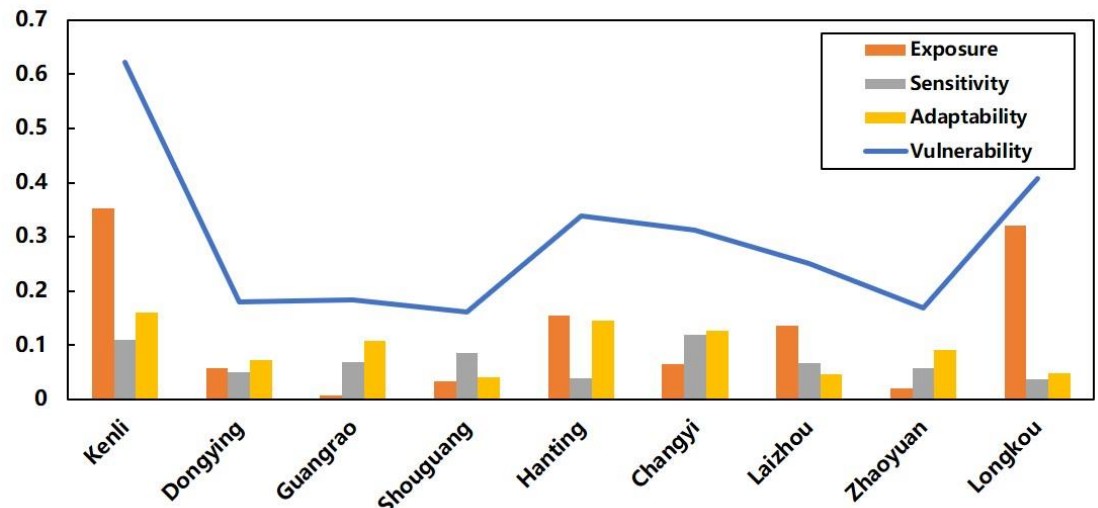

**Figure 6.** Estimation of vulnerability to storm surge in the coastal area of Laizhou Bay.

The exposure index was calculated for areas with different exposure values as indicators, and the results revealed that the value of the exposure index ranged from 0.008–0.353, with a higher exposure-index value indicating higher exposure to storm surges. Among the nine assessment units, the exposures of the Kenli District and Longkou City were significantly higher than those of the other areas. A large number of oil fields and culture ponds are located in the coastal area of the Kenli District. Moreover, the Kenli District has more water bodies, low terrain, and a gentle slope, resulting in high natural exposure. Hence, the exposure of the Kenli District is high. The high exposure of Longkou City was attributed to the relatively significant recent urban development. Ports and built-up areas are located along the coastline. Therefore, even though the elevation of the Longkou District

is relatively high and the slope is steeper than that of the Kenli District, it is still highly vulnerable. The vulnerability levels of the Hanting District and Laizhou City are also relatively high. The natural environment exposures in these two areas are equivalent. There are more salt fields and industrial land areas in the Hanting District. The large-scale Weifang Port is also located in the Hanting District coastal zone. There are also many salt fields and ports in the coastal zone of Laizhou City, and there are a relatively large number of urban and rural residential areas in Laizhou City. Therefore, although the vulnerability levels in the Hanting District and Laizhou City are lower than those in the Kenli District and Longkou City, they are still considerable. The exposure of Guangrao County was the lowest, since its coastline is very short, and there are fewer disaster-bearing bodies located in this coastal area than in other areas. From the fine-scale land-use analysis, the local exposure of an assessment unit can be obtained. Therefore, decision-makers can easily identify regions with high exposure values in an assessment unit. In addition, all disaster-bearing bodies can be included in the overall exposure assessment, making the exposure assessment more accurate and comprehensive.

Five indicators were used to calculate the sensitivity index, which was found to vary from 0.037–0.120. The lower range of the sensitivity index indicates less sensitivity and decreased vulnerability to storm surge. The results revealed that the Kenli District and Changyi City have higher sensitivity. The main reason for this is that among the five sensitivity indicators, the weight of the ratio of fishery products to GDP is the largest, and the ratios of fishery products to GDP in the Kenli District and Changyi City are much higher than in other regions. At the same time, the second-highest indicator weight is the percentage of females, and the percentages of females in the Kenli District and Changyi City rank third and first in the region, respectively. Therefore, the sensitivity in these two regions is high. Conversely, the sensitivity of the Hanting District is the lowest, and it is no surprise that the statistical results of the two indicators of the ratio of fishery products to GDP and percentage of females in the Hanting District are the lowest of the nine regions in Laizhou Bay.

The adaptability index was calculated using six variables, and was found to range from 0.041–0.160. Lower values of the adaptability index indicate greater adaptability, and reduced vulnerability to storm surge. As shown in Figure 5, the adaptability values of the Kenli District, Hanting District, and Changyi City were substantially higher than those of other counties and cities. The level of economic development in the Hanting District is relatively low, and its values of the six indicators used to assess adaptability were low, resulting in this district exhibiting the lowest adaptability. In contrast, the social and economic development levels of Longkou City surpassed those of other counties and cities, and its values of the six indicators were relatively high, leading to this area displaying the highest adaptability of the study area. It can be deduced that the overall levels of social and economic development are closely related to adaptability. In addition, Laizhou City's health care level was higher than that of other areas, increasing its adaptability to storm surge.

Finally, the exposure, sensitivity, and adaptability indices were combined in order to obtain the vulnerability value of each area in Laizhou Bay. The vulnerability values ranging from [0,1] were divided into Levels I, II, III, and IV. Figure 7a is the vulnerability level distribution of the coastal zone of Laizhou Bay. Among the nine regions, the Kenli District has the highest vulnerability level, reaching Level II vulnerability, mainly due to its prominent exposure, sensitivity, and adaptability index values. Level III vulnerability areas include the Hanting District, Changyi City, Laizhou City, and Longkou City, while the Dongying District, Guangrao County, Shouguang City, and Zhaoyuan City have the lowest vulnerability, Level IV.

Observation data for the maximum storm water increase in the shore sections of Laizhou Bay were obtained from the "Bohai Sea Storm Surges and Disasters" and expressed as the storm surge hazard index. The hazard levels were converted, with the corresponding relationships listed in Table 11.

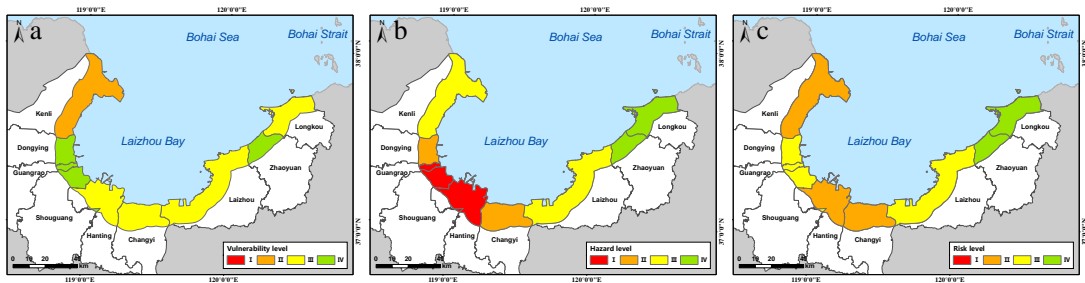

**Figure 7.** Assessment results for Laizhou Bay: (**a**) Vulnerability; (**b**) Hazard; (**c**) Risk.

**Table 11.** Corresponding relationship between the storm surge hazard and storm water increase.

| Hazard Level | Storm Water Increase (cm) |
| --- | --- |
| Level I | ≥350 |
| Level II | [300–350) |
| Level III | [250–300) |
| Level IV | <250 |

The storm surge hazard in each district following the conversion is illustrated in Figure 7b. According to the recorded data, the extreme value of storm water increase along the coast of Laizhou Bay has ranged from 150–350 cm, and the maximum water increase has occurred in the southern portion of the bay in the southwest direction, with the intensity of the water increase gradually decreasing from the southern end to the mouth of the bay. Among the nine assessment units, Guangrao County, Shouguang City, and the Hanting District exhibited maximum water increases of ≥350 cm, which represents the highest hazard level. In contrast, the extreme water increases of storms in Zhaoyuan City and Longkou City in the northeast part of the bay ranged from 150–200 cm, corresponding to the lowest hazard level, IV.

Based on the comprehensive vulnerability and hazard levels, according to the corresponding relationship in Table 7, a storm surge risk assessment was performed on the coastal zone of Laizhou Bay. The results are illustrated in Figure 7c.

As can be seen from the Figure 7, the Kenli District, Hanting District, and Changyi City exhibit the highest risks, reaching Level II. The Kenli District includes a large number of highly-exposed oilfields and other industrial land. At the same time, the "fisheries output value" in the sensitivity index is high, and the overall socioeconomic level is relatively low, resulting in high vulnerability, with the hazard level of this district reaching Level III. According to the corresponding relationship in Table 7, the risk level of the Kenli District was determined to be Level II. The three districts with the highest risk were found to be Guangrao County, Shouguang City, and the Hanting District, all of which reached Level I hazard, although Guangrao County and Shouguang City had lower vulnerability values. Therefore, in the risk assessment results of these three high-hazard areas, only the Hanting District, which has relatively high vulnerability, reached a Level II risk. Due to its particular vulnerability and high hazard level, Changyi City's final risk evaluation result also reached Level II. In other areas, the risk levels of Dongying City, Guangrao County, Shouguang City, and Laizhou City were all Level III, while those of Zhaoyuan City and Longkou City were Level IV.

The risk assessment results for the Laizhou Bay area are the superposition of the vulnerabilities and hazards. According to the results of the vulnerability assessment for the Laizhou Bay area, vulnerability is highly related to exposure, followed by adaptability, while the sensitivity index has less impact than the first two indices. The coastal zone of the Kenli District in the Laizhou Bay area is the most vulnerable, mainly due to the existence of a large number of industrial land-use types such as salt fields, low-lying regions, gentle slope areas, and numerous water systems. In terms of both land-use type and natural conditions, it has a high degree of exposure. At the same time, the adaptability index of the Kenli District is also low. According to Table 9, the medical and economic conditions of the Kenli

District are relatively low, resulting in this area having the highest regional vulnerability in Laizhou Bay. The hazards in the Laizhou Bay area exhibit an obvious geographical pattern, i.e., high at the bottom of the bay and low at the mouth of the bay. The hazard levels of Guangrao, Shouguang, and Hanting are the highest in Laizhou Bay. The risk distribution characteristics of storm surge in the Laizhou Bay area are as follows: Hanting and Changyi, which are mainly affected by high hazard levels, and the Kenli District, which is highly vulnerable to disasters, are the high-risk areas, while the remainder are relatively low-risk areas.

## 5. Discussion

### 5.1. Uncertainty Analysis

The uncertainty in this study was examined in terms of four aspects: (1) the uncertainty of the data; (2) the uncertainty of the index selection; (3) the uncertainty of the weight setting; and (4) the uncertainty of the disaster risk.

Data uncertainty: the data used in this study included GF-2 remote sensing image data, DEM data, vulnerability value data of disaster-bearing bodies from the State Oceanic Administration of China, social and economic statistical data, and historical storm water increase data used in risk assessment. The land-use data of the GF-2 remote sensing images after object-oriented interpretation were checked by manual visual interpretation, which reduced the uncertainty in the interpretation process to the lowest possible level. There were some differences, however, between the edges of the patches after segmentation and the edges of the actual objects on the remote sensing images, resulting in uncertainty. There was also uncertainty in the DEM data. The spatial resolution of the DEM used in this study was 30 m, which can reflect certain terrain conditions. However, we still thinks it is not precise enough, and that it may have affected the evaluation results. The basic exposure value of a hazard-bearing body in this study was the corresponding exposure value of a hazard-bearing body in the guidelines of the State Oceanic Administration of China, although this value is a guide, and therefore, is not the only correct exposure value of the hazard-bearing body type. It is inappropriate to use the same exposure value for the coastal cities of China, which would generate uncertainty. Social and economic statistics are determined every year. In this study, we utilized the 2017 statistics, which were temporally synchronized with the remote sensing image imaging time, although there are undoubtedly some errors and uncertainties in the statistics. The social statistical data, such as the population proportions, were based on the national census data of China, which are obtained every 10 years. The social statistical data in this study were from 2010, which produced differences in the data synchronization, and then generated relatively large uncertainties in the evaluation results. Finally, the storm surge data were based on the station observation data over the years. The uncertainty of these data was low and the results were relatively reliable.

Indicator uncertainty: when selecting the sensitivity and adaptivity indicators in this study, we first conducted relevant literature research [21,60–62], then organized discussions with experts, and finally selected the indicators by considering the data availability. The indicator selection in this investigation was mainly reflected by three aspects: population composition, economic situation, and medical level. We believes that when storm surges and other sudden disasters occur, the elderly, children, and the less educated are more vulnerable to the threats from these disasters. At the same time, families with relatively small amounts of economic resources have relatively poor tolerance to natural disasters such as storm surge, while the medical level of a region is directly related to the response ability to sudden disasters. In addition, since these indicators can be obtained from census data or urban statistical yearbooks, the data are readily available. Only using the indicators in this study, however, may not be comprehensive or biased, and will thus have a certain impact on the evaluation of vulnerability, resulting in uncertainty. Methods for choosing more appropriate indicators will be an important research direction in the follow-up study.

Weight setting uncertainty: because the indicators in this study were not completely consistent with those of other studies, there was no article that could be referred to when evaluating the importance of each indicator. Therefore, the weight of each index (i.e., the relative importance of each index) was determined by way of expert scoring. Although basing the weight settings on expert experience can reduce their deviations, it cannot be denied that in such a complex system, expert experience may also differ from the actual situation, thus affecting the evaluation results. In order to reduce uncertainties of this type, we can only accumulate experience via actual cases or similar natural disasters, and then adjust the weights and indicators.

Hazard uncertainty: in this study, the observation data of storm water increases in the ocean region of the experimental area were used to directly classify and assign the hazard of each district and county. This is a very simplified risk assessment method, and there was no simulation of a flood plain or other situations. The main reason is that the author considers the flood plain phenomenon of storm surge to be very complex, in which flood control measures and drainage facilities, as well as the influence of humans or other factors, such as water increase, land cover type, seawall, and so on, all need to be considered. Even if the simulation is successfully carried out, there may be large deviations. Additionally, the storm surge data used in this study did not consider the superposition of sea-level rise caused by climate change, since without flood plain simulation, sea-level rise would have the same impact on each district and county in the study area. Moreover, because the main focus of this study was vulnerability assessment, as well as the framework of refined storm surge vulnerability and risk assessment based on remote sensing data, there was no detailed analysis of disaster risk, and the final risk assessment results were uncertain. In a follow-up study, we will add a more detailed disaster hazard assessment method. In that project, the sea level rise caused by climate change will not be ignored, since it needs to be analyzed along with the storm surge in order to determine the maximum surge and inundation range.

### 5.2. Comparison with Other Research

Since there are no other previous research results in this study area, and the data used for the methods proposed in this study are different from the data on which other methods in the references are based, direct application of other methods may lead to inaccurate evaluation results due to data discrepancies, and direct comparison of the results is of little significance. In this study, we made a comparison in terms of the methods in order to reveal the different concerns of this investigation and those of other works.

The comparative method was the approach used by Paprotny [6]. Paprotny carried out the risk assessment of coastal zone flooding in Poland. Thus, their study was based on the land-use and DEM data for the coastal zone of Poland. In their project, risk assessment was divided into three facets: hazard, exposure, and vulnerability. Hazard mainly included the scope of the inundation area and the water increase caused by sea level rise (SLR) and storm surge. The coastal zone terrain information was obtained from DEM data, and the inundation area was calculated. The maximum increment of 5 m was the superposition of SLR and the 1000-year storm surge. Exposure was determined by land-use type and corresponding market values. Vulnerability was the superposition of land-use type exposure and disaster probability. The risk assessment in our study was the combination of hazard and vulnerability. Hazard was classified according to the maximum storm water increase, while vulnerability was the combination of exposure, sensitivity, and adaptability. The exposure was mainly based on the qualitative assessment of different land-use types by the State Oceanic Administration of China. At the same time, our study focused on increasing the impacts of the natural environment on the exposure, using the three indices of elevation, slope, and water system distance to comprehensively determine the exposure. Sensitivity and adaptability were determined from different social indices.

The differences between our study and Paprotny's study are mainly reflected by the following: (1) Different forms of exposure evaluation. Paprotny used market values to realize the evaluation of land-use type exposure, but the experimental area of our study was located in China, in which the

economic development levels of different cities vary considerably, even among coastal cities, and the land use-type and economic development levels in the North and South are also different. Therefore, it was not appropriate to use the average market values of land-use types for the entire country or coastal cities, or the market values of specific urban land-use types to evaluate the exposure. In future research, however, the market values of local urban land-use types can be taken into account as an evaluation index of the exposure. (2) The DEM data were used in different ways. In our study, the DEM data were used to extract slope and elevation information, combining the natural environment and exposure, while Paprotny utilized the DEM data to determine the flooded area. (3) The hazard evaluation methods were different. The hazard evaluation in the Paprotny study relied on the DEM to simulate the flooded area. In this study, the hazard assessment was based on the historical storm surge data of the study area and counties, which allowed the direct classification of the hazard level for each area and county, without simulating the flood plain. The main reason for this was previously explained in the related section of the uncertainty analysis. The primary purpose of this study was to propose a framework of refined vulnerability and risk assessment. Therefore, a very simplified hazard assessment method was adopted, and more detailed simulation results obtained by other hazard assessment methods could also be applied to the framework proposed in this study as part of the hazards.

The main purpose of this study was to propose a framework, through the use of remote-sensing satellite data, for obtaining fine-scale (patch scale) land-use data. These fine-scale land-use data could then be scaled up, and vulnerability and risk assessment at different scales could be performed, in order to play a guiding role in the early warning and emergency response of different levels of local governments. The example employed in this study was to combine the socioeconomic data and disaster risk data of districts and counties as a unit to realize the vulnerability and risk assessment of districts and counties. In the specific application, one can select other levels of data, such as municipal and provincial statistical data, for a more macroscale assessment. If there are more appropriate methods for the specific vulnerability assessment methods, risk assessment methods, and indicators used in the framework, they could also be replaced accordingly.

## 6. Conclusions

In this study, a remote sensing-based fine-scale storm surge vulnerability and risk assessment model was proposed, and the assessment of storm surge vulnerability and risk was carried out in Laizhou Bay, Shandong Province, China. The main innovations of the model are as follows:

(1) Based on high-precision remote sensing images, the vulnerability and risk assessment with adjustable scale was realized. Through the interpretation of high-precision remote-sensing images, the land-use patch objects were obtained. On this basis, by combining the socioeconomic data of different statistical scales, the vulnerability and risk assessment of storm surge could be realized on the scale of the block, county, and city. For different scale data, the combination ability was found to be strong and the operability high.

(2) In the vulnerability assessment, the impact of the natural environment on the exposure of the disaster-bearing body was added. Since the disaster-bearing body is located in the natural environment, the impact of the natural environment on the body needs to be considered. In this study, the impact of the natural environment was abstracted into three indicators: terrain, slope, and water system distance. By combining the exposure of the disaster-bearing body with the natural environment indicators, the exposure of the disaster-bearing body was obtained, taking into consideration the natural environment factors, yielding storm surge vulnerability and risk assessment results that were closer to reality.

Given the precise and reliable data as well as the comprehensive considerations of the proposed storm surge vulnerability and risk assessment model, this more comprehensive approach can provide relatively reliable assessment results for local management departments. In addition, the large-area, high-precision characteristics of the remote sensing observations can provide a reliable data source for storm surge risk assessment. In this study, land-use data obtained from remote sensing data were used

to achieve a refined vulnerability assessment. Moreover, the potential of remote sensing does not stop there. Marine observations and inversion data obtained by remote sensing can also support the hazard assessment of storm surges, and they can even realize the dynamic risk assessment of marine disasters, such as storm surges, a goal that warrants further study.

**Author Contributions:** Conceptualization, Y.L., C.L., X.Y., and Z.W.; Data curation, Y.L., Z.W., and B.L.; Formal analysis, Y.L. and Z.W.; Funding acquisition, X.Y.; Investigation, Y.L., C.L., Z.W., and B.L.; Methodology, Y.L. and C.L.; Project administration, X.Y.; Resources, X.Y.; Software, Y.L. and C.L.; Supervision, X.Y.; Validation, Y.L., X.Y., Z.W., and B.L.; Visualization, Y.L.; Writing—original draft, Y.L. and C.L.; Writing—review and editing, Y.L. All authors have read and agreed to the published version of the manuscript.

**Funding:** This research was funded by the CAS Earth Big Data Science Project (Grant No. XDA19060202), the National Key Research and Development Program of China (Grant No. 2016YFC1402003), the National Science Foundation of China (Grant No. 41671436), and the Innovation Project of LREIS (Grant No. O88RAA01YA).

**Conflicts of Interest:** The authors declare no conflict of interest.

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
