# Peer review of "Fine-Scale Coastal Storm Surge Disaster Vulnerability and Risk Assessment Model: A Case Study of Laizhou Bay, China"

_remotesensing, doi:10.3390/rs12081301_

Round 1

Reviewer 1 Report

The paper entitled “Fine-scale coastal storm surge disaster vulnerability and risk assessment model: A case study of Laizhou Bay” is an interesting paper. The topic of the paper is suitable for the “Remote Sensing” journal. In my opinion the manuscript requires modification and revision I order to be improved and become accepted for publication.

Here are some comments and suggestion which should be taken into consideration by the authors:

  1. What is the border of the study area depicted in Figure 1? Did the authors consider as study area the coastal area from the coastline up to 10 km inland? Or the criterion was a maximum elevation? Please explain. The impacts of storm surges had impact on manmade constructions and/or environmentally important elements at a distance of 10 km inland?
  2. The figure of the study area map should include an inst map showing the location of the study area within China. The study area map should also provide elevation data. What is the topography of the area? It is very important for the reader to know the relief. Additionally a map showing the main cities, the population density and the land use types (prior their categorization into exposure values) could help the reader to understand the conditions within the study area.
  3. A record of the storm surge events that have affect the study area should be provided (at the Study Area section) and a frequency of the storm surge events should be given providing an idea about the negative results and the affected localities. Maybe a list of the most severe storm surge events and the corresponding losses (regarding human life and material) could help the reader to understand if the study area is a storm surge prone region. Somewhere at the results it is mentioned that a e-level rise of 350 cm due to storm surge is the limit of the higher hazard level. The study area is that low-lying that a sea-level rise of 350 cm affects 10 km inland?
  4. At the methodology section it should be clear from the beginning what is the main administration unit that all estimations and assessments are made eg municipality? Municipal community? Country? District?
  5. What was the resolution of the DEM used to estimate “environment” parameters? What was the source of this DEM? Topographic maps? If yes at what scale? Of what contour interval? Satellite images?
  6. At the methodology section the authors does not mention at all in what way the land use types have been obtained from satellite images. What was the procedure? What is the accuracy of the method they applied?
  7. In most similar vulnerability assessment approaches as a demographic parameter is considered the “percentage of population under age 5” and not the “percentage of population under age 15”. I propose to the authors to change this sensitivity parameter according to my comment otherwise they should explain why they consider as “helpless” teenagers at the age of 15.
  8. In my opinion the results are given in a very complicated way and the comments on the results are just descriptive. There are not clear comments about the geographic distribution of the vulnerability and risk and what are the most important factors for this distribution.
  9. In my opinion the authors should add some references of coastal vulnerability assessment methods:

Boruff, B.; Emrich, C.; Cutter, S.L. Erosion hazard vulnerability of US coastal countries. J. Coast. Res. 2005, 21, 932–942.

Cutter, S.L.; Mitchell, J.T.; Scott, M.S. Revealing the vulnerability of people and places: A case study of Georgetown country, South Carolina. Ann. Assoc. Am. Geogr. 2000, 90, 713–737.

Karymbalis, E.; Chalkias, C.; Ferentinou, M.; Chalkias, G.; Magklara, M. Assessment of the Sensitivity of Salamina and Elafonissos islands to Sea-level Rise. J. Coast. Res. 2014, 70, 378–384.

Author Response

Dear reviewers and editors,

Thank you very much for your comments on our manuscript (ID: remotesensing-764633). Based on your comments, we have carefully revised the manuscript and polished the language by consulting a professional English editing service.

The responses to each comment are as the following in red and the original comments are in black. We sincerely hope the revised version could encourage your acceptance of this paper.

We look forward to your comments.

Yours sincerely,

Authors of the manuscript remotesensing-764633

To facilitate your review of our revisions, the following are our point-by-point responses.

Note: The line number of the manuscript mentioned in our response refers to the line number of the revised manuscript in word version, which you can download from the system, rather than the pdf version generated by the system or the pdf version with your comments uploaded by you.

Author Responses:

Response to Reviewer 1 Comments:

The paper entitled “Fine-scale coastal storm surge disaster vulnerability and risk assessment model: A case study of Laizhou Bay” is an interesting paper. The topic of the paper is suitable for the “Remote Sensing” journal. In my opinion the manuscript requires modification and revision I order to be improved and become accepted for publication.

Here are some comments and suggestion which should be taken into consideration by the authors:

Point 1: What is the border of the study area depicted in Figure 1? Did the authors consider as study area the coastal area from the coastline up to 10 km inland? Or the criterion was a maximum elevation? Please explain. The impacts of storm surges had impact on manmade constructions and/or environmentally important elements at a distance of 10 km inland?

Response 1:

Thanks for your insightful questions. The scope of our study area was the buffer zone from the coastline to 10 km inland, which was determined based on the description in the literature [50], i.e., when a severe storm surge disaster occurs, it causes seawater to flow back along the river and spread 5–10 km inland. Therefore, we selected 10 km as the maximum impact range of storm surge disasters. In the text, we also included supplementary explanations. Please refer to line 130–133 in the revised manuscript.

Point 2: The figure of the study area map should include an inst map showing the location of the study area within China. The study area map should also provide elevation data. What is the topography of the area? It is very important for the reader to know the relief. Additionally a map showing the main cities, the population density and the land use types (prior their categorization into exposure values) could help the reader to understand the conditions within the study area.

Response 2:

Thanks for your suggestions and questions. We have added a map showing the location of the study area in Figure 1, as well as maps depicting the area’s topography and population information. At the same time, we added the land-use results after remote sensing image interpretation of the study area (Figure 3). Please refer to line 137 and line 334–343 in the revised manuscript.

Point 3: A record of the storm surge events that have affect the study area should be provided (at the Study Area section) and a frequency of the storm surge events should be given providing an idea about the negative results and the affected localities. Maybe a list of the most severe storm surge events and the corresponding losses (regarding human life and material) could help the reader to understand if the study area is a storm surge prone region. Somewhere at the results it is mentioned that a e-level rise of 350 cm due to storm surge is the limit of the higher hazard level. The study area is that low-lying that a sea-level rise of 350 cm affects 10 km inland?

Response 3:

Thanks for your comments. We have added information on the frequency of storm surge disasters in the study area, and also included some of the data on storm surge disasters and damage in this region (Table 2). Please refer to line 139–146 in the revised manuscript.

The reason for the selection of the 10-km range has been clarified in the response to your first question. Namely, it was primarily due to the fact that the longest distance of tidal water flowing up the river in the historical record was 10 km. Hence, the 10-km range was not selected based on topography.

Point 4: At the methodology section it should be clear from the beginning what is the main administration unit that all estimations and assessments are made eg municipality? Municipal community? Country? District?

Response 4:

Thanks for your helpful questions. The smallest unit in this study was the patch-scale land-use datum. By combining the statistical data of different scales, the vulnerability and risk assessment of block units, district units, or city units could be achieved. Since the example of Laizhou Bay in this study used district-county–level statistical data, the evaluation was conducted in district-county–level units. The relevant content has been added in the Methodology section. Please refer to line 213–219 in the revised manuscript.

Point 5: What was the resolution of the DEM used to estimate “environment” parameters? What was the source of this DEM? Topographic maps? If yes at what scale? Of what contour interval? Satellite images?

Response 5:

Thanks for your helpful questions. This study utilized 30-m–resolution SRTM DEM data. The relevant content has been added in the Materials section, and we examine the uncertainty of the DEM data in the Discussion section.

Please refer to line 166–169 and line 520–523 in the revised manuscript.

Point 6: At the methodology section the authors does not mention at all in what way the land use types have been obtained from satellite images. What was the procedure? What is the accuracy of the method they applied?

Response 6:

Thanks for your comment and questions. In this study, the interpretation of remote sensing images used a combination of object-oriented and visual procedures. We have added the explanation to the Methodology section and also included the image interpretation results (Figure 3).

Please refer to line 177–198 and line 334–343 in the revised manuscript.

Point 7: In most similar vulnerability assessment approaches as a demographic parameter is considered the “percentage of population under age 5” and not the “percentage of population under age 15”. I propose to the authors to change this sensitivity parameter according to my comment otherwise they should explain why they consider as “helpless” teenagers at the age of 15.

Response 7:

Thanks for your comments and suggestions. The reason for choosing this indicator in this study is that in China, children start school at the age of 6 and receive 9 years of compulsory education. Therefore, compared with adults, it may be more difficult for young people under the age of 15 to take appropriate refuge measures when confronted with sudden disasters. At the same time, because the index of the population under 5 years of age is included in the index of the population under 15 years of age, we chose the latter. In the corresponding portion of the text, we have added the relevant explanatory content. Please refer to line 294 (Table 5) in the revised manuscript.

Point 8: In my opinion the results are given in a very complicated way and the comments on the results are just descriptive. There are not clear comments about the geographic distribution of the vulnerability and risk and what are the most important factors for this distribution.

Response 8:

Thanks for pointing this out. In the final portion of the Results section, we have added the geographical distribution characteristics of vulnerability, hazard, and risk, and analyzed the factors that affected their distribution results. Please refer to line 492–507 in the revised manuscript.

Point 9: In my opinion the authors should add some references of coastal vulnerability assessment methods:

Boruff, B.; Emrich, C.; Cutter, S.L. Erosion hazard vulnerability of US coastal countries. J. Coast. Res. 2005, 21, 932–942.

Cutter, S.L.; Mitchell, J.T.; Scott, M.S. Revealing the vulnerability of people and places: A case study of Georgetown country, South Carolina. Ann. Assoc. Am. Geogr. 2000, 90, 713–737.

Karymbalis, E.; Chalkias, C.; Ferentinou, M.; Chalkias, G.; Magklara, M. Assessment of the Sensitivity of Salamina and Elafonissos islands to Sea-level Rise. J. Coast. Res. 2014, 70, 378–384.

Response 9:

Thank you very much for the above reference for our work. This is of great help to our work. We have added references to the relevant literature, i.e., [21], [22], and [45]. Please refer to line 56 and line 70 in the revised manuscript.

At the same time, we also added some other reference related to our study. Ultimately, the number of references increased to 63.

Thank you again for your serious and responsible comments.

Reviewer 2 Report

The article is interesting, and the subject is worthy of research. However, the execution of the article and the research as well as presentation itself requires some important improvements to proceed with its publication in the journal in my opinion. For this I advise a major revision to the authors in the following points:

There are many oversights and mismatches with the format of the journal in the material execution. Here are some of them:  unjustified typography changes especially in tables, missing authors contribution, lack of line numeration, etc.

In a number of cases, you had facts, information, ideas or methods that were not your own, which had no in-text citation. It is very important to stick a reference at the end of the paragraph. I noticed, in many lines or paragraphs there are no references cited (it is hard to precisely show each case as there is no line numeration prepared in the manuscript).

It is unclear to me what is the significance and innovation of the work discussed in the paper, compared to other similar works. Is it only the fact that this is the first study addressing this issue in analysed area? Authors state: “compared with the conventional method using only statistical data, the evaluation results have higher credibility” but in discussion I find no comparison to any other works.

In the figure 1 I miss a general area representation. I advise to add one general figure representing  the described area within a bigger perspective as not all international readers might be familiar with the study area or provinces and city names only.

In introduction authors miss number of recent articles that deals with coastal flooding generally and flood risk assessment especially with its implications that suggests that the literature review is still to be done. Please add more positions concerning flooding in general and some examples of works dealing with different types of floods not necessarily only those connected directly with storm surge. Consider works like Salman, AM (2018), Dandapat, K (2017), Paprotny D and Terefenko P (2017) Zischg, AP (2018), Paprtony D (2020) among others. I suggest to rewrite this part considering suggested newest works. Citing around 25 positions totally in the manuscript (17 in Introduction) also looks really poor and one more time suggests that the literature review has to be redone.

What about climate change? Please keep in mind that climate changes will not only lead to precipitation increase or sea level rise, but may also result in changes in storminess, which affects the frequency of extreme events. In this case it should be at least mentioned about some possibilities of adaptation measures. For now the adaptation aspects are poorly developed. While analyzing flood assessment for risk planning it is necessary to describe it in more details.

I miss a more detail description of remote sensing aspects. Section 3.1 presents and describes briefly RS data used for analysis. Though this one short section it is not enough to fulfill the “remote sensing” idea of the work. Without this part the manuscript would better fit some other MDPI journals like “Water” for example. As it was submitted to “Remote Sensing” please provide more information about the data, processes, analysis of land use extraction. This part has a fundamental importance for fulfilling the aims and scope of the journal.

How did the population indicators were selected. Was there an analysis performed that showed specific indicators such as percentage of females, percentage of population under age 15, and percentage of population aged 65 and above as well as education aspects as a crucial ones? If not those were chosen arbitrary. Please explain your choice. If they were chosen randomly authors will need to perform additional analysis. Some historical datasets could be helpful.

It was not a good idea to merge result and discussion section as authors does not refer their results directly to the values achieved by others. Similarly as in introduction I still miss a real discussion to a worldwide literature. Actually it seems that merging those allowed to skip the discussion part at all.

The discussion should shows very clearly the advantages of proposed methodology in relation both to other existing models and a real dataset. Later the conclusions should be  self-contained so that the  reader who jumps straight into the Conclusion will be able to understand them.

Finally what is very important it is unclear to me what is the significance and innovation of the work discussed in the paper, compared to other similar works. Actually I miss a clear statement preferably presented in Interlocution what is the main aim of the article. Further this main aim should be addressed in Conclusion section.

Author Response

Dear reviewers and editors,

Thank you very much for your comments on our manuscript (ID: remotesensing-764633). Based on your comments, we have carefully revised the manuscript and polished the language by consulting a professional English editing service.

The responses to each comment are as the following in red and the original comments are in black. We sincerely hope the revised version could encourage your acceptance of this paper.

We look forward to your comments.

Yours sincerely,

Authors of the manuscript remotesensing-764633

To facilitate your review of our revisions, the following are our point-by-point responses.

Note: The line number of the manuscript mentioned in our response refers to the line number of the revised manuscript in word version, which you can download from the system, rather than the pdf version generated by the system or the pdf version with your comments uploaded by you.

Author Responses:

Response to Reviewer 2 Comments:

The article is interesting, and the subject is worthy of research. However, the execution of the article and the research as well as presentation itself requires some important improvements to proceed with its publication in the journal in my opinion. For this I advise a major revision to the authors in the following points: 

Point 1: There are many oversights and mismatches with the format of the journal in the material execution. Here are some of them: unjustified typography changes especially in tables, missing authors contribution, lack of line numeration, etc.

Response 1:

Thanks for pointing this out. We have revised the manuscript based on the issues you raised, including adding line numbers, adjusting the format of the tables, adding the author contributions as well as the conflicts of interests, and so on.

Point 2: In a number of cases, you had facts, information, ideas or methods that were not your own, which had no in-text citation. It is very important to stick a reference at the end of the paragraph. I noticed, in many lines or paragraphs there are no references cited (it is hard to precisely show each case as there is no line numeration prepared in the manuscript).

Response 2:

Thanks for pointing this out. We have added references, especially for non-original information, in the Study Area and Materials and Methodology sections.

Please refer to line 43, line 46, line 50, line 54, line 58, line 119, line 150, line 172–174, and line 222 in the revised manuscript.

Point 3: It is unclear to me what is the significance and innovation of the work discussed in the paper, compared to other similar works. Is it only the fact that this is the first study addressing this issue in analysed area? Authors state: “compared with the conventional method using only statistical data, the evaluation results have higher credibility” but in discussion I find no comparison to any other works.

Response 3:

Thanks for pointing this out. We have explicitly listed the significance and innovative aspects of this study in the Introduction, Discussion, and Conclusions sections, and supplemented a comparison with other methods in the Discussion section. Due to the lack of comparative research results in this study area, and the different methods and data used in this study in contrast to other studies, we only compared this study with other studies in terms of methods in the Discussion section, and also revealed the focus and innovation of this study compared with other studies.

Please refer to line 108–112, line 622–632, and line 638–651 in the revised manuscript.

Point 4: In the figure 1 I miss a general area representation. I advise to add one general figure representing  the described area within a bigger perspective as not all international readers might be familiar with the study area or provinces and city names only.

Response 4:

Thanks for pointing this out. In Figure 1, we have added a map showing the location of the study area, as well as maps depicting the terrain and population distribution in the study area.

Please refer to line 137 in the revised manuscript.

Point 5: In introduction authors miss number of recent articles that deals with coastal flooding generally and flood risk assessment especially with its implications that suggests that the literature review is still to be done. Please add more positions concerning flooding in general and some examples of works dealing with different types of floods not necessarily only those connected directly with storm surge. Consider works like Salman, AM (2018), Dandapat, K (2017), Paprotny D and Terefenko P (2017) Zischg, AP (2018), Paprtony D (2020) among others. I suggest to rewrite this part considering suggested newest works. Citing around 25 positions totally in the manuscript (17 in Introduction) also looks really poor and one more time suggests that the literature review has to be redone.

Response 5:

Thank you very much for the above reference for our work. This is of great help to our work. We have strengthened our introduction section based on the above reference and some other reference, and rewrote the related portions of the Introduction. Please refer to line 63–81 in the revised manuscript. Ultimately, the number of references increased to 63.

Point 6: What about climate change? Please keep in mind that climate changes will not only lead to precipitation increase or sea level rise, but may also result in changes in storminess, which affects the frequency of extreme events. In this case it should be at least mentioned about some possibilities of adaptation measures. For now the adaptation aspects are poorly developed. While analyzing flood assessment for risk planning it is necessary to describe it in more details.

Response 6:

Thanks for your comments. We have added some references and discuss climate change briefly in the Discussion section. Please refer to line 68–81 and line 560–576 in the revised manuscript.

The main purpose of this study was to propose a framework model, focusing on the combination of remote sensing data in the study of storm surge vulnerability and risk assessment, and to realize a vulnerability and risk assessment model that can be adjusted to a particular scale, in order to provide reference for the disaster management and protection by local governments. Therefore, this study simplified the relevant aspects of disaster hazard assessment. In the Discussion section, we list the shortcomings of this study. At the same time, the impact of climate change on disasters is multifaceted. We will add more information in the follow-up study in order to upgrade the model of this research.

Point 7: I miss a more detail description of remote sensing aspects. Section 3.1 presents and describes briefly RS data used for analysis. Though this one short section it is not enough to fulfill the “remote sensing” idea of the work. Without this part the manuscript would better fit some other MDPI journals like “Water” for example. As it was submitted to “Remote Sensing” please provide more information about the data, processes, analysis of land use extraction. This part has a fundamental importance for fulfilling the aims and scope of the journal.

Response 7:

Thanks for pointing this out. We have enhanced the remote sensing explanation. We have also added a description of the advantages of remote sensing to the Introduction. In the Materials section, we have added the relevant satellite parameter information for the data used in this study (Table 2). In the Methodology section, we have supplemented the specific methods and steps of the remote sensing interpretation utilized in this study. In the Results section, we have added the land use results (Figure 3).

Please refer to line 93–107, line 163 and line 177–198 in the revised manuscript.

Point 8: How did the population indicators were selected. Was there an analysis performed that showed specific indicators such as percentage of females, percentage of population under age 15, and percentage of population aged 65 and above as well as education aspects as a crucial ones? If not those were chosen arbitrary. Please explain your choice. If they were chosen randomly authors will need to perform additional analysis. Some historical datasets could be helpful.

Response 8:

Thanks for your questions and comments. For the indicator selection, we convened experts and scholars to discuss which indicators to select, evaluated and screened all of the indicators we obtained from the expert scoring, and finally determined the indicators listed in this manuscript, i.e., females, the elderly, and children, all of whom exhibit greater sensitivity to disasters. We also discussed the uncertainties of the indicator selection in the Discussion section.

Please refer to line 289–293 and line 537–550 in the revised manuscript.

Point 9: It was not a good idea to merge result and discussion section as authors does not refer their results directly to the values achieved by others. Similarly as in introduction I still miss a real discussion to a worldwide literature. Actually it seems that merging those allowed to skip the discussion part at all.

Response 9:

Thanks for your comments. We have expanded the Discussion section, in which we now examine the uncertainty of the research model, as well as the comparison between this study and other studies, and reveal the focus and innovation of this study.

Please refer to line 508–632 in the revised manuscript.

Point 10: The discussion should shows very clearly the advantages of proposed methodology in relation both to other existing models and a real dataset. Later the conclusions should be  self-contained so that the  reader who jumps straight into the Conclusion will be able to understand them.

Response 10:

Thanks for your comments. We compare the proposed methodology with other methods in the Discussion section, then reveal the focus and innovation of this study, and combine and demonstrate the innovative aspects of this project in the Conclusions section.

Please refer to line 577–651 in the revised manuscript.

Point 11: Finally what is very important it is unclear to me what is the significance and innovation of the work discussed in the paper, compared to other similar works. Actually I miss a clear statement preferably presented in Interlocution what is the main aim of the article. Further this main aim should be addressed in Conclusion section.

Response 11:

Thank you very much for the valuable comments. We have added the main purpose and innovation of this study to the Introduction, Discussion, and Conclusion sections, in which we present clear descriptions of the above content. The focus and innovation of this study are primarily reflected in 2 achievements: (1) A fine-scale storm surge vulnerability and risk assessment model based on remote sensing images was proposed, which can be combined with statistical data of different scales to achieve adjustable vulnerability and risk assessment at a particular scale and provide disaster prevention and control information for local management departments. (2) The impacts of natural environmental factors were combined with the exposure of the disaster-bearing body, so as to conduct a more complete disaster vulnerability assessment.

Please refer to line 108–112, line 622–632, and line 638–651 in the revised manuscript.

Thank you again for your serious and responsible comments.

Reviewer 3 Report

The manuscript is generally ok but needs some improvement before publication. 

Please find my detailed comments in the attached annotated PDF. 

Author Response

Dear reviewers and editors,

Thank you very much for your comments on our manuscript (ID: remotesensing-764633). Based on your comments, we have carefully revised the manuscript and polished the language by consulting a professional English editing service.

The responses to each comment are as the following in red and the original comments are in black. We sincerely hope the revised version could encourage your acceptance of this paper.

We look forward to your comments.

Yours sincerely,

Authors of the manuscript remotesensing-764633

To facilitate your review of our revisions, the following are our point-by-point responses.

Note: The line number of the manuscript mentioned in our response refers to the line number of the revised manuscript in word version, which you can download from the system, rather than the pdf version generated by the system or the pdf version with your comments uploaded by you.

Author Responses:

Response to Reviewer 3 Comments:

The manuscript is generally ok but needs some improvement before publication.

Please find my detailed comments in the attached annotated PDF.

Point 1: Please define specific objectives for your research here.

Response 1:

Thank you very much for the valuable comments. To make our works motivation clearer, we have added the research objectives of this study in the last portion of the Introduction, mainly including: “(1) A storm surge vulnerability and risk assessment method combining remote sensing images is proposed; (2) When vulnerability assessment is performed, natural environmental factors are added to the exposure assessment; (3) Taking the coastal area of Laizhou Bay, Shandong Province, China as the research area, a storm surge vulnerability and risk assessment is conducted.”

Please refer to line 108-112 in the revised manuscript. And it will be helpful for readers to understand the motivation of our work. Thank you again for pointing this out.

Point 2: please show that where is the location of your study area in China? all the readership may do not know the location of your study area in china.

Response 2:

Thanks for your suggestion. In Figure 1, we have added a map showing the location of the study area, as well as maps depicting the terrain and population distribution in the study area. Please refer to line 137 in the revised manuscript.

Point 3: Please provide a table showing the technical characteristic of GF-2 sensor.

Response 3:

Thanks for your suggestion. We have added Table 2, which presents the GF-2 sensor information. Please refer to line 163 (Table 2) in the revised manuscript.

Point 4: please separate discussion from results

Response 4:

Thanks for your comments. We have added the Discussion section, in which we examine the uncertainties of the research model, as well as the comparison between this study and other studies, and also reveal the focus and innovative aspects of this study.

Please refer to line 508-651 in the revised manuscript.

Point 5: please increase the resolution of the image (Figure 3 and Figure 6 in the original manuscript).

Response 5:  

Thanks for your suggestion. We have re-used a clearer EMF-format image and replaced it in order to improve the resolution. Please refer to Figure 4 and Figure 7 in the revised manuscript.

Point 6: Some updated references are required for your manuscript.

Response 6:  

Thank you for pointing that out. We have added the updated references. Ultimately, the number of references increased to 63.

Thank you again for your serious and responsible comments.

Round 2

Reviewer 2 Report

Congratulations to authors.

I find all my remarks carefully analyzed and answered.

I accept the manuscript in current version.